# Experimental Study on the Preparation of Ultra-Fine Brass Tube Electrodes by Ultrasonic Vibration

**DOI:** 10.3390/mi14061234

**Published:** 2023-06-12

**Authors:** Hanlin Yu, Yugang Zhao, Zhihao Li, Chuang Zhao, Shuo Meng, Yu Tang, Xiajunyu Zhang, Guangxin Liu

**Affiliations:** School of Mechanical Engineering, Shandong University of Technology, Zibo 255049, China

**Keywords:** ultrasonic vibration processing, ultra-fine brass tube electrode, brass tube with core, copper tube electrode processing

## Abstract

In order to automatically process ultra-fine copper tube electrodes, this study proposes a new method of ultrasonic vibration processing of ultra-fine copper tube, analyzes its processing principle, designs a new set of experimental processing equipment and completes the processing of 1.206 mm inner diameter, 1.276 mm outer diameter with core brass tube. Not only can the copper tube be completed with core decoring, the surface of the processed brass tube electrode also has good integrity. The effect of each machining parameter on the surface roughness of the electrode after machining was investigated by a single-factor experiment and the optimal machining effect was achieved under the conditions of machining gap 0.1 mm, ultrasonic amplitude 0.186 mm, table feed speed 6 mm/min, tube rotation speed 1000 r/min and reciprocating machining two times. The surface roughness was reduced from 1.21 μm before machining to 0.11 μm, and the residual pits, scratches and oxide layer on the surface were completely removed, which greatly improved the surface quality of the brass tube electrode and prolonged its service life.

## 1. Introduction

Microporous structures are used more and more widely in the fields of molding, medical devices, aerospace, etc. In the field of aero-engines, the turbine blades of aero-engines are subjected to the maximum thermal load during operation and the working environment is very harsh, so the air-film hole cooling technology is a key technology for protecting the turbine components from overheating [1]. The aero-turbine engine airfoil holes are characterized by large number, small apertures and high hardness of turbine blades, which are extremely difficult to machine by traditional methods [2]. High-speed EDM provides a practical solution to this problem because of its advantages of fast machining speed, suitable for multi-hole machining, small heat-affected zone, no surface micro-cracking, and wide range of processed materials [3]. Compared with conventional EDM, it uses a tubular electrode, during which a high-pressure water-based working fluid is added inside the tubular electrode to force chip evacuation, while the electrode rotates at high speed to make small holes more uniformly machined [4]. In this process, the copper tube electrode is crucial, as its diameter will determine the size of the inner diameter of the processed small hole, and the surface quality of the electrode will determine the surface quality of the inner wall of the processed small hole. Meanwhile, the copper tube electrode is a consumable part [5], and a large amount of high-quality copper tube electrodes are required for batch processing of small holes; thus, the current urgent problem to be solved is how to produce high-quality ultra-fine copper tube electrodes with high efficiency, especially with automatic processing.

In the field of EDM cylindrical electrode machining, there are many methods: for example, using the method of electrical discharge drilling (EDD), ultra-fine WC-6% Co cylindrical electrodes with an aspect ratio of 45 can be machined [6]; additionally, using the method of multi-EDM grinding, cylindrical electrodes with a length of 0.6 mm and a diameter of 20 μm can be machined [7]. Other methods such as Block electrode discharge grinding (BEDG) and Electrochemical Etching [8] are also widely used, and the processing of cylindrical electrodes is also developing in the direction of ultrafine, high aspect ratio and high-quality electrode materials. In contrast, compared with the processing of cylindrical electrodes, the preparation methods of ultrafine tube electrodes have rarely been studied, and there are few studies in the field of automated processing of copper tube electrodes with high aspect ratios. At present, the widely used production process of ultra-fine copper tube is as follows: (1) insert a steel core inside the thicker copper tube blank, and compress the copper tube blank by mechanical rolling or drawing to obtain a finer copper tube with a core; (2) stress relieving annealing; (3) roll the copper tube with a core by mechanical knocking to separate the inner wall of the copper tube from the steel core, and withdraw the steel core to obtain a finer copper tube that is decored; (4) repeat the above process until the desired outer diameter is obtained. The most important step here is the decoring. Mechanical knocking decoring efficiency is low, the yield rate is low, the preparation area is large, it is difficult to achieve automation, etc. The decoring force in the processing of copper tube electrode is often very large and can only be done by machines, so we innovatively try to use ultrasonic vibration to replace mechanical knocking to achieve the decoring of copper tubes with cores. The decoring force is significantly reduced after processing, and the steel core can even be easily removed by hand; at the same time, the surface of the processed copper tube electrode has a better integrity, and the processing effect is good. Ultrasound is widely used in welding, milling, grinding, drilling, laser processing and other fields [9,10,11,12], but mainly for auxiliary manufacturing. Direct use of ultrasound for mechanical processing is not yet common in the field of machining. There is a small amount of research on processing metal materials directly using ultrasonic vibration—for example, ultrasonic impact treatment [13], which is closer to the principle of shot peening, can make use of the advantages of high ultrasonic vibration frequency and low force to make the surface of the workpiece undergo small plastic deformation and surface grain refinement, thus enhancing the surface quality and improving the mechanical properties [14]. Ultrasonic impact treatment has the advantages of high ultrasonic vibration frequency, low force, nano-scale deformation, strengthening the surface of metal materials, significantly improving the surface integrity of metal materials, generating beneficial residual stress on the surface, improving the mechanical properties of the surface, etc. It is considered one of the most promising technologies in the cold treatment of metal materials [15]. Therefore, we tried to use ultrasonic vibration instead of mechanical knocking to decore brass tubes with cores, which could not only result in automatic and high-efficiency production of ultra-fine copper tube electrodes but could also improve the surface quality of the processed copper tube electrodes and prolong the service life of the electrodes.

This study proposes and designs an ultrasonic processing machine tool for processing ultra-fine brass tube electrodes, summarizes the relevant processing parameters affecting the surface roughness of brass tube electrodes according to the mechanism of ultrasonic processing, designs a single-factor experiment to investigate the effect of each experimental parameter on the surface roughness of processed brass tube electrodes, and obtains good processing results.

## 2. Experimental Equipment and Materials

Figure 1 shows the developed experimental equipment for ultrasonic vibration preparation of ultra-fine brass tube electrodes. The mechanical structure of the equipment mainly consists of three parts: the ultrasonic processing part, the tube clamping and rotation part, and the processing platform transverse movement part. The ultrasonic processing part mainly consists of an ultrasonic generator and machining gap adjustment device. The ultrasonic generator, amplitude rod, transducer and ultrasonic forging head constitute the ultrasonic generator. The machining gap adjustment device consists of an AC servo motor and ball screw module, which provides the ultrasonic vibration required for processing and adjusts the machining gap in the process. The tube clamping and rotation part consists of a precision chuck, two AC servo motors and a tube tensioning device. The brass tube with core is clamped on the two AC servo motors through the precision chuck to provide the rotational movement required for processing. The lateral movement part of the processing platform consists of a stepper motor, a synchronous belt linear module and a lateral movement sliding table. In addition, the rear fixed frame can be moved on the lateral movement sliding table, and by adjusting the distance between it and the front fixed frame, tube workpieces of different lengths can be processed. The interface of the control system is written in Delphi language, and the control system program is written in VC++6.0. The control of the AC servo motor is installed symmetrically on the same axis. The stepping motor in the horizontal direction and the AC servo motor in the vertical direction can be realized simultaneously, which makes the tube rotate and reciprocate, and the ultrasonic forging head moves linearly perpendicular to the processing plane. The performance of the experimental equipment for ultrasonic vibration preparation of ultra-fine copper tube electrodes is shown in Table 1.

The outer diameter of the brass tube with core and the diameter of the steel core (inner diameter) were measured at three different locations using a spiral micrometer and then averaged. The outer diameter of the brass tube with core used in this experiment was 1.276 mm and the diameter of the middle steel core was 1.206 mm; the material of the outer brass tube was brass and its composition is shown in Table 2 [16]. The material of the internal steel core was grade 45 steel, and the length of the processed tube was 600–900 mm.

## 3. Introduction of Processing Principle

The principle of ultrasonic processing of brass tube electrodes is shown in Figure 2 below. After roller pressing, both sides of the brass tube with core are fixed on two servo motors through precision collets to provide rotary motion. The guiding mechanism prevents the brass tube with core from rotating in the process of lateral jump and also provides alignment and guidance. The lower part of the brass tube with core is the forging block. The upper part of the ultrasonic generator, through the transducer mechanism to amplify the vibration, and finally through the ultrasonic forging head to the brass tube with core is processed by the ultrasonic forging head, and the brass tube with core does transverse reciprocating motion during the processing. The processing process and deformation principle is as follows: After the roller press, the brass tube and steel core are closely combined and difficult to separate, as shown in Figure 3a. The processing gap h exists between the brass tube with core and the ultrasonic forging head before the processing starts, and the ultrasonic forging head will ultrasonically strike the surface of the brass tube with core after the processing starts; a single strike is shown in Figure 3b. At this time, under pressure, the brass tube undergoes a small plastic deformation. A gap is generated between the core and the external brass tube due to the high rotational speed of the brass tube with core and the high frequency of ultrasonic vibration. The final deformation of the brass tube shape cross-section is close to a circle, and the external brass tube and the iron core appear to have a uniform gap. The brass tube and the iron core separation force are significantly reduced and the results are shown in Figure 3c. At the same time, due to the ultrasonic processing of the ultrasonic forging head on the external brass tube, forging numerous times makes the surface reinforced. Some defects, scratches and rust on the original surface are partially or completely removed and the surface quality is greatly improved, which can significantly improve the service life of the processed brass tube electrode.

The essence of ultrasonic processing is to use the advantages of small macroscopic force and high processing frequency of ultrasonic vibration to replace the disadvantages of large macroscopic force and low processing frequency of mechanical knocking. However, its microscopic deformation mechanism, lattice changes and microscopic processing mechanism, such as processing resonance brought about by ultrasonic vibration, are yet to be investigated in depth. Based on a large number of experiments performed so far, we make the following analysis of the influence of each parameter on the processing results.

Figure 4 is a schematic diagram of the effect of the amplitude of the ultrasonic forging head and the machining gap on the surface processing of brass tubes during the machining process. Without considering the radial runout and vibration of the brass tube with core during the actual processing, the ideal situation during processing is that when the ultrasonic forging head vibration reaches the limit position, it can process the surface of the brass tube with core and make the inner wall of the brass tube deformed with the internal core to create a gap. When the ultrasonic forging head vibration reaches the limit position and cannot be processed to the surface of the brass tube electrode, or it can be processed to the outer surface of the brass tube electrode but not enough to make the inner surface of the brass tube plastic deformed with the core to create a gap, it is difficult to extract the core after processing; this results in failed processing. When the amplitude of the ultrasonic forging head is too large or the processing gap is too small, so that the external brass tube has too much plastic deformation, under the joint action of resonance and brass tube rotation the brass tube electrode outside the core will directly break and pop out. This situation is to be avoided in the processing process; thus, a reasonable choice of ultrasonic amplitude and processing gap is the key factor in successful processing.

Since the vibration frequency generated by the ultrasonic generator is fixed at about 20 kHz, the uniformity of the brass tube surface processing is determined by the tube rotation speed, the table feed speed and the number of times reciprocal machining is performed. As the brass tube rotates around the axis and the two movements of the lateral feeding of the brass tube, it is decided that the trajectory of the ultrasonic forging head processing on the electrode surface of the brass tube during the one-way lateral feeding is a spiral line, and the pitch of this spiral line is decided by the two parameters of the brass tube rotational speed and the table feed speed. When the table feed speed is fixed, the larger the rotational speed of the brass tube, the smaller the pitch of the spiral line, the more dense the processing trajectory, and the more uniform the processing, but because the vibration frequency is fixed, the rotational speed is too large. This will lead to a reduction in the number of times of processing per unit area of the brass tube surface, the plastic deformation of the brass tube will not be obvious, and finally, the core and the brass tube cannot be separated, resulting in processing failure. When the rotational speed is too small, the number of times of processing under the unit area of the brass tube surface is too high, and this will lead to poor surface quality after processing; in serious cases, this may lead to the direct shattering of the external brass tube, and the impact of the table feed speed on the processing results is the same. As the processing trajectory is a spiral line when the table is fed in one direction, the surface of the brass tube may not be completely processed, so the processing process table is reciprocating. When the table returns to the movement, the pitch is unchanged, but the rotation direction is opposite and the processing trajectory will be crossed, so that the surface of the brass tube can be fully and evenly processed. Therefore, a reasonable choice of tube rotation speed, table feed speed and the number of times that reciprocal machining is performed will directly determine the quality of the surface of the brass tube electrode after processing.

## 4. Experimental Results and Discussion

We adjusted the rear fixed frame to a suitable position, fixed the brass tube with core to be processed on the precision collet of the servo motor and made the brass tube with core pass through the guiding device, adjusting the tensioning handwheel to tighten the brass tube with core to prevent radial runout during processing. We then input the prepared CNC program into the computer, applied a little lubricant at the contact between the guiding mechanism and the brass tube with core, started the power supply and started processing. Each experiment processed a 20 mm long brass tube electrode sample. The processed sample was wiped clean, the appropriate amount of anhydrous ethanol was added to the beaker, followed by the brass tube electrode sample. The beaker was placed into the ultrasonic cleaner to ultrasonically clean the processed brass tube electrode to avoid stains on the outer surface of the tube for subsequent observation and measurement. The surface roughness of each selected position of the tube was measured using a DSX1000 3D digital microscope (DSX1000, OLYMPUS, Tokyo, Japan) at an interval of 3 mm, and the average value was calculated to discuss the effect of each processing parameter on the surface roughness of the processed brass tube electrode.

In a large number of pre-experiments and processing experiments on copper tubes, we found a reasonable parameter interval that ensures that the steel core inside the copper tube can be removed smoothly and that the electrodes of the copper tubes are not shattered during processing. After expanding this parameter interval, we obtained the experimental parameter interval as shown in Table 3, and this was used to conduct single-factor experiments.

### 4.1. Influence of Machining Gap

To investigate the effect of machining gap on the surface roughness of brass tube electrodes, a total of five sets of experiments were conducted in 0.05 mm increments with other parameters fixed, and the experimental parameters are shown in Table 4. The effect of different machining gaps on surface roughness is shown in Figure 5.

The line graph shows that when the machining gap is 0.1 mm, the surface roughness of the brass tube electrode after machining is 0.25 μm. Other machining gap conditions, although they can successfully decore, the surface roughness is above 0.3 μm, which is not optimal; Figure 6a is the 3D digital micrograph of the surface of the brass tube electrode when the machining gap is 0.05 mm. We can obviously see that there are many small depressions on the surface of the brass tube electrode, which is because the processing gap is too small, resulting in too much deformation on the surface of the brass tube electrode and resulting in the surface hardening off. At this time, if the machining gap is further reduced, this will directly lead to the brass tube electrode breaking during processing and processing failure. As the processing gap increases, the surface roughness decreases and reaches the lowest value of 0.25 μm when the processing gap is 0.1 mm. Additionally, the surface roughness increases with the increase in the processing gap; when the processing gap exceeds 0.2 mm, the decoring force of the brass tube electrode with core obviously increases, which is due to the large gap. This leads to most areas of the surface of the brass tube electrode not being processed; thus, the surface of the brass tube electrode will still have more defects, then continuing to increase the processing gap will mean the surface of the brass tube electrode cannot be processed. Figure 6b shows the 3D digital micrograph of the surface of the brass tube electrode when the processing gap is 0.2 mm; it is obvious that there are still very many defects on the surface, most of the oxide layer is still there, and the surface metal luster is dull.

### 4.2. Effect of Ultrasonic Amplitude

To investigate the effect of ultrasonic amplitude on the surface roughness of brass tube electrodes, an experiment was conducted in increments of 0.012 mm with other parameters fixed. The experimental parameters are shown in Table 5, and the effect of different ultrasonic amplitudes on surface roughness is shown in Figure 7.

Figure 7 shows that when the ultrasonic amplitude is too small, the surface processing of the brass tube electrode is not complete; the surface roughness is reduced compared with the original surface, but it is not optimal. Figure 8a is the 3D digital micrograph of the surface of the brass tube electrode at the ultrasonic amplitude of 0.162 mm. The surface of the brass tube electrode still has some unremoved patina, the surface metal luster of the electrode is not obvious, and the residual defects still exist; thus, processing is not complete. With increasing ultrasonic amplitude, the surface roughness of the brass tube electrode obviously decreases, and the surface roughness reaches an optimal 0.25 μm when the ultrasonic amplitude is 0.186 mm. When the amplitude of ultrasonic vibration reaches 0.21 mm, the knocking of ultrasonic vibration sometimes also causes the processed brass tube electrode to break, which directly leads to processing failure. Figure 8b shows the 3D digital micrograph of the brass tube electrode surface at the ultrasonic amplitude of 0.21 mm, which shows that there are many craters on the electrode surface caused by excessive deformation.

### 4.3. Influence of Table Feed Speed

To investigate the effect of table feed speed on the surface roughness of brass tube electrodes, an experiment was conducted in increments of 1.5 mm/min with other parameters fixed. The experimental parameters are shown in Table 6, and the effect of different table feed speeds on surface roughness is shown in Figure 9.

It can be seen from Figure 9 that as the table feed speed increases, the surface roughness of the brass tube electrode starts to decrease, and reaches the minimum value of 0.15 μm when the table feed speed reaches 6 mm/min. The surface roughness of the brass tube electrode increases when the table feed speed continues to increase, which is due to the fact that the faster the table feed speed is, the less the number of knocks per unit area the brass tube electrode surface is subjected to. When the table feed speed is too fast, the electrode surface is not completely processed, and the surface roughness of the electrode increases. When the feed speed reaches 10.5 mm/min or more, the cored brass tube will resonate because of the processing vibration, so the brass tube electrode breaks at the crest, and the diameter of the fracture obviously increases. When the table feed speed is too slow, the number of knocks per unit area of the electrode surface is too high, and the defects will increase. At the same time, there will be a great residual stress on the surface, and the surface of the processed workpiece will have craters visible to the naked eye when it is left for more than two days. The lower the table feed speed is, the more obvious this phenomenon is, and even fracture will occur when the feed speed is lower than 1.5 mm/min.

### 4.4. Effect of Tube Rotation Speed

To investigate the effect of tube rotation speed on the surface roughness of brass tube electrodes, an experiment was conducted in 200 r/min increments with other parameters fixed. The experimental parameters are shown in Table 7, and the effect of different tube rotation speeds on surface roughness is shown in Figure 10.

As can be seen from the figure, with increasing tube rotation speed, the surface roughness of the brass tube electrode first decreases and then increases, and the minimum surface roughness of the electrode is 0.11 μm at 1000 r/min. The influence of tube rotation speed on the surface roughness of the electrode after processing is essentially the same as for table feed speed, which is to influence the processing by changing the number of knocks per unit area of the brass tube electrode surface. However, because of the inevitable radial runout of the cored brass tube in the rotation process, when the tube rotation speed is too high—although the surface of the brass tube electrode is not over-processed—there will still be a sudden fracture of the brass tube in the process. Using a DSX1000 3D digital microscope to observe the fractured brass tube electrode, it was found that the surface processing was not complete, and the surface scratches, pits, and so on still existed. This is due to the speed being too high; the brass tube electrode radial runout also increases, so that a part of the brass tube electrode surface processing gap is reduced or even touches the ultrasonic forging head, which leads to the part being shattered and fracturing. When the tube rotation speed is too low, the brass tube electrode surface unit is over-processed; this produces processing defects and its surface roughness will also increase.

### 4.5. Influence of the Number of Times Reciprocating Machining Is Performed

The effects of performing reciprocating machining a different number of times on surface roughness are shown in Figure 11, and the experimental parameters are shown in Table 8.

In the process of ultrasonic vibration processing of brass tube electrodes, the one-way processing trajectory of the ultrasonic forging head on the surface of the brass tube electrode is a spiral line, and two processing trajectories will cross in one reciprocating motion. When the number of times reciprocating machining is performed is increased, the processing of the electrode surface will be more uniform. However, too much reciprocating machining will lead to a longer processing time, lower efficiency, and surface over-processing and defects. Therefore, the amount of reciprocating machining should be selected reasonably, taking into account both the final electrode surface processing effect and the processing efficiency. As can be seen from Figure 11, when reciprocating machining is performed two times, the surface roughness of the brass tube electrode is 0.11 μm. When reciprocating machining is performed three times, the surface roughness of the brass tube electrode is 0.10 μm, which is not a big improvement, but the processing time is extended by half. Thus, the best processing effect can be achieved when reciprocating machining is performed two times.

### 4.6. Comparison before and after Processing

Figure 12a shows the 3D digital micrograph of the surface of the brass tube with core of 1.206 mm inner diameter and 1.276 mm outer diameter before ultrasonic machining. It is obvious that the surface of the brass tube has been oxidized, and there are a lot of defects such as pits and scratches caused by mechanical rolling or drawing. The surface roughness is 1.21 μm, which will shorten the service life of the brass tube electrode and seriously affect the surface quality of the workpiece after EDM. Figure 12b shows the 3D digital micrograph of the surface of the brass tube electrode after ultrasonic machining, with the machining parameters of 0.1 mm gap, 0.186 mm ultrasonic amplitude, 6 mm/min table feed speed, 1000 r/min tube rotation speed, and reciprocating machining two times. The oxide layer on the surface of the brass tube electrode was completely removed, and the mechanical defects produced by the previous process were also removed. The surface organization was dense and bright, and the surface roughness reached 0.11 μm, which achieved good processing results.

## 5. Conclusions


The innovative introduction of ultrasonic vibration into the processing of brass tube electrodes has achieved good processing results, which can ensure good surface integrity of the processed brass tube electrode surface under the condition that decoring can be completed smoothly, and at the same time solved the problem where decoring the ultra-fine brass tube electrodes in batches with high efficiency and automation is difficult with the traditional mechanical knocking method. This proves that the introduction of ultrasonic vibration into the processing of brass tube electrodes is practical and feasible.The principle of ultrasonic vibration processing of brass tube electrodes was analyzed, and various processing parameters affecting the processing results were introduced. The influence of each experimental parameter on the surface roughness of the processed electrode was investigated through single-factor experiments. It was verified that with the experimental parameters of 0.1 mm machining gap, 0.186 mm ultrasonic amplitude, 6 mm/min table feed speed, 1000 r/min tube rotation speed and reciprocating machining two times, the obtained brass tube electrode has the best surface integrity and the surface roughness value can reach 0.11 μm.


## Figures and Tables

**Figure 1 micromachines-14-01234-f001:**
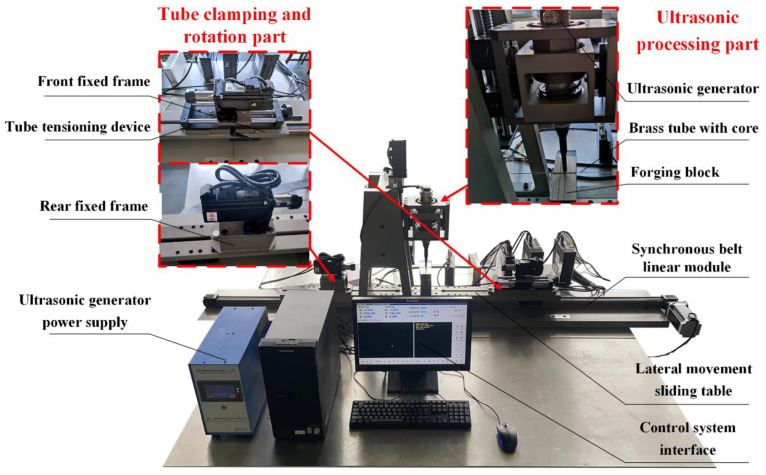
Ultrasonic vibration preparation of ultra-fine brass tube electrodes equipment.

**Figure 2 micromachines-14-01234-f002:**
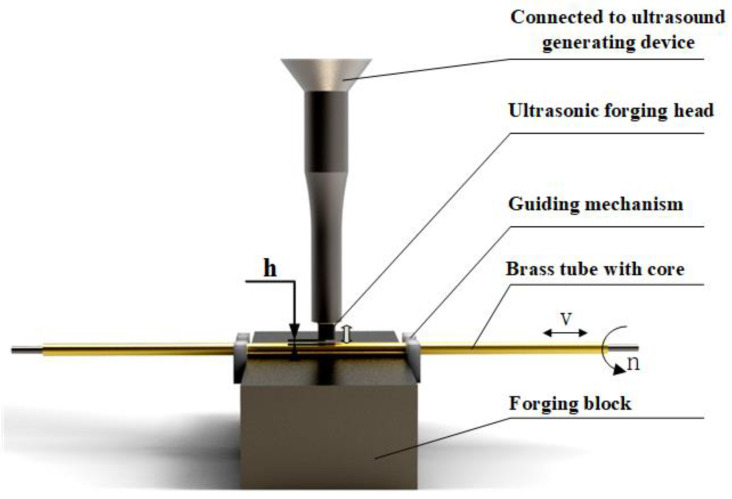
Ultrasonic processing brass tube electrode schematic.

**Figure 3 micromachines-14-01234-f003:**
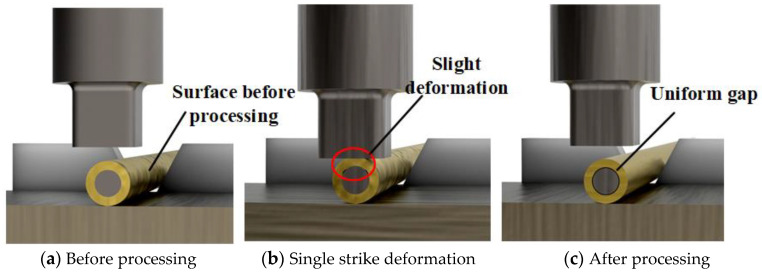
Ultrasonic processing brass tube electrode deformation principle.

**Figure 4 micromachines-14-01234-f004:**
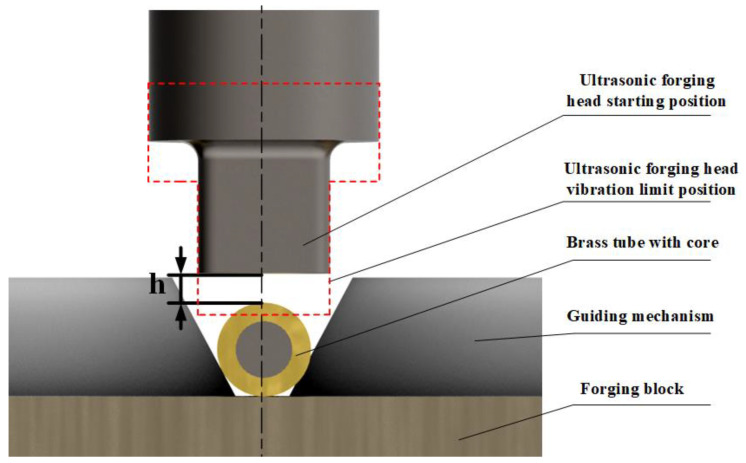
Effect of ultrasonic forging head amplitude and processing gap on the surface processing of brass tubes.

**Figure 5 micromachines-14-01234-f005:**
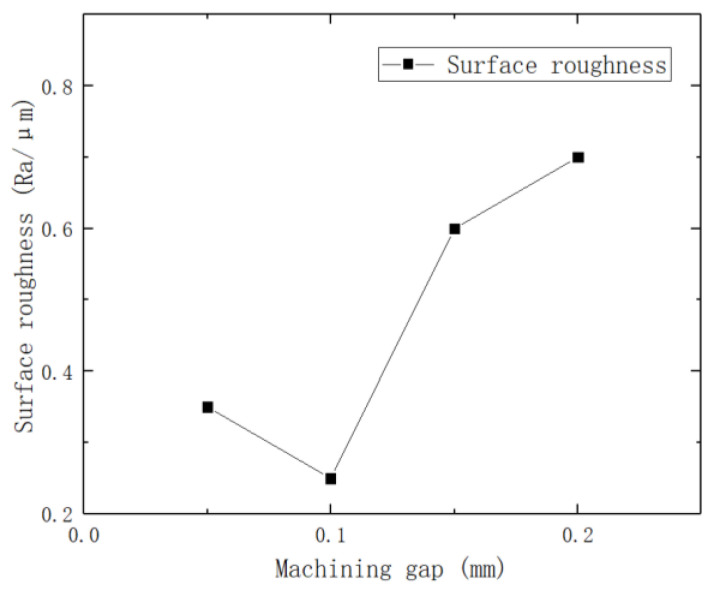
Effect of different machining gaps on surface roughness.

**Figure 6 micromachines-14-01234-f006:**
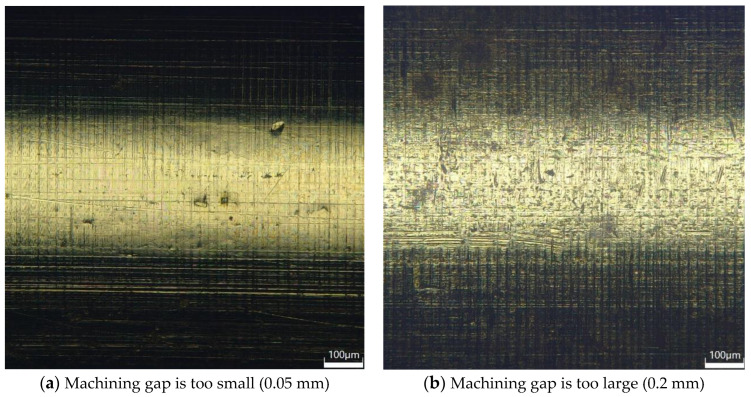
3D digital micrograph of electrode surface of brass tube with different machining gaps.

**Figure 7 micromachines-14-01234-f007:**
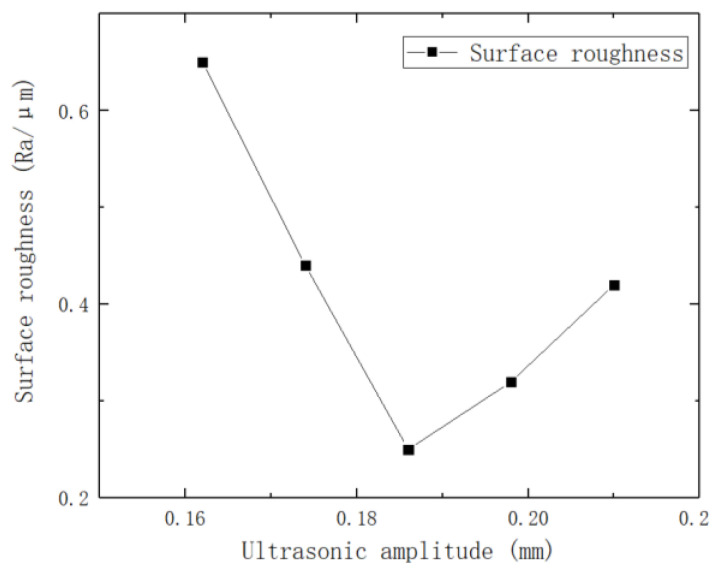
Effect of different ultrasonic amplitudes on surface roughness.

**Figure 8 micromachines-14-01234-f008:**
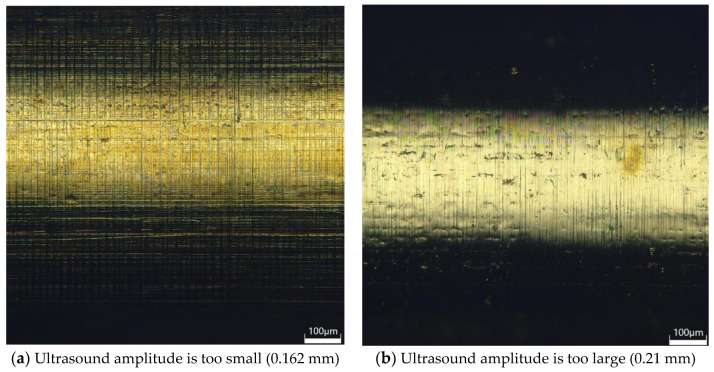
3D digital micrographs of the electrode surfaces of brass tubes with different ultrasonic amplitudes.

**Figure 9 micromachines-14-01234-f009:**
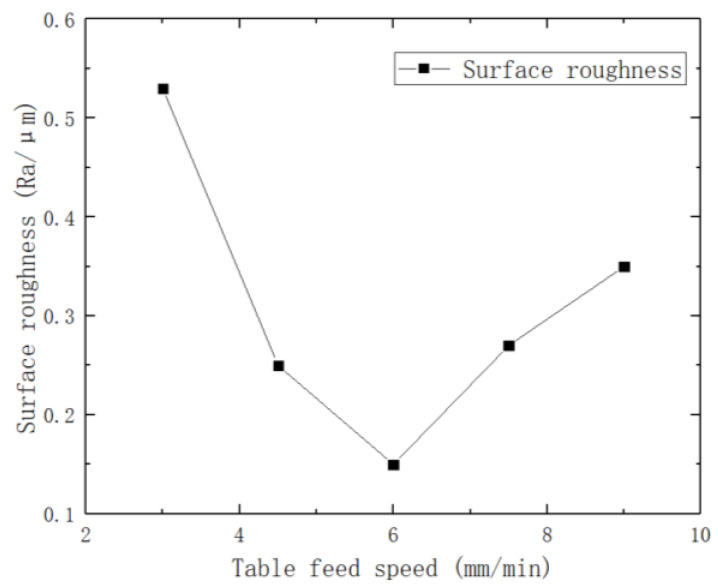
Influence of different table feed speeds on surface roughness.

**Figure 10 micromachines-14-01234-f010:**
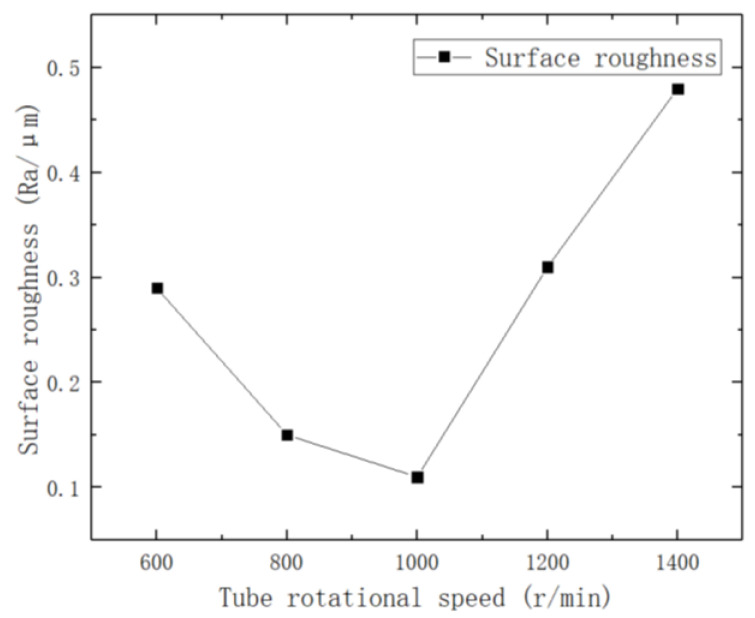
Effect of tube rotation speed on surface roughness.

**Figure 11 micromachines-14-01234-f011:**
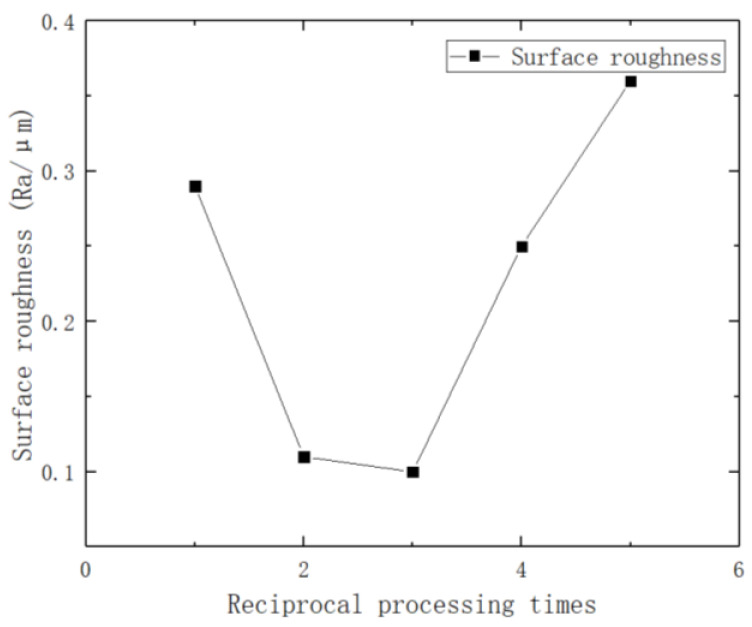
Effect of performing reciprocal machining a different number of times on surface roughness.

**Figure 12 micromachines-14-01234-f012:**
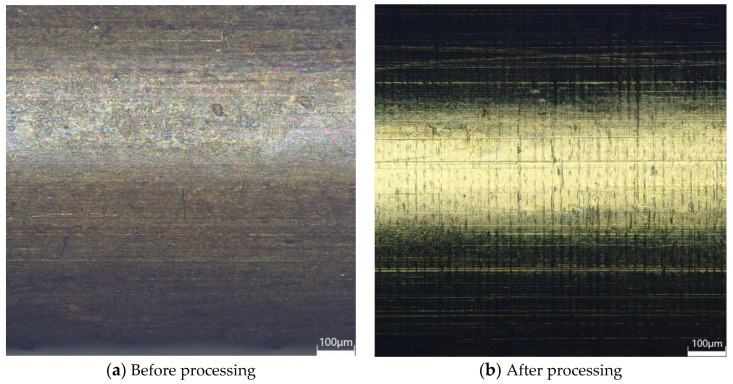
Comparison of 3D digital micrograph of the surface of electrode before and after processing.

**Table 1 micromachines-14-01234-t001:** Performance of ultrasonic percussion processing experimental equipment for brass tube electrodes.

Performance	Parameter Range
Processable tube diameter range (mm)	0.5~2
Machining stroke (mm)	50~1500
Tube rotation speed (r/min)	0~3000
Table lateral movement speed (mm/min)	0~300

**Table 2 micromachines-14-01234-t002:** Brass composition—element content.

Element	Copper	Zinc	Iron	Niobium and Molybdenum	Lead
Weight (%)	63.21	35.96	0.35	0.18	0.12

**Table 3 micromachines-14-01234-t003:** Experimental parameters and intervals.

Parameter Name	Experimental Parameter Interval
Machining gap	0~0.2 mm
Ultrasonic amplitude	0.162~0.21 mm
Table feed speed	3~9 mm/min
Tube rotation speed	600~1400 r/min
Number of reciprocal machining	1~5

**Table 4 micromachines-14-01234-t004:** Table of experimental parameters for different machining gaps.

Parameter Name	Experimental Parameters
Machining gap	0~0.2 mm
Ultrasonic amplitude	0.186 mm
Table feed speed	4.5 mm/min
Tube rotation speed	800 r/min
Number of reciprocal machining	2

**Table 5 micromachines-14-01234-t005:** Table of experimental parameters of different ultrasonic amplitudes.

Parameter Name	Experimental Param Eters
Machining gap	0.1 mm
Ultrasonic amplitude	0.162~0.21 mm
Table feed speed	4.5 mm/min
Tube rotation speed	800 r/min
Number of reciprocal machining	2

**Table 6 micromachines-14-01234-t006:** Table of experimental parameters for different table feed speeds.

Parameter Name	Experimental Parameters
Machining gap	0.1 mm
Ultrasonic amplitude	0.186 mm
Table feed speed	3~9 mm/min
Tube rotation speed	800 r/min
Number of reciprocal machining	2

**Table 7 micromachines-14-01234-t007:** Table of experimental parameters for different tube rotation speeds.

Parameter Name	Experimental Parameters
Machining gap	0.1 mm
Ultrasonic amplitude	0.186 mm
Table feed speed	6 mm/min
Tube rotation speed	600~1400 r/min
Number of reciprocal machining	2

**Table 8 micromachines-14-01234-t008:** Table of experimental parameters for different number of reciprocating processing.

Parameter Name	Experimental Parameters
Machining gap	0.1 mm
Ultrasonic amplitude	0.186 mm
Table feed speed	6 mm/min
Tube rotation speed	1000 r/min
Number of reciprocal machining	1~5

## Data Availability

Not applicable.

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
