# Peer review of "Experimental Study on the Preparation of Ultra-Fine Brass Tube Electrodes by Ultrasonic Vibration"

_micromachines, 2023, doi:10.3390/mi14061234_

Round 1

Reviewer 1 Report

1.The abstract is recommended to be modified to avoid the use of "success" and other similar words. Consider modifying the Abstract “…and successfully completed the processing of 1.206mm inner diameter….”. It could not be made unsuccessfully what you just did and you did not present any debate over its possibility. So, delete the unnecessary word "successfully", as it adds to jargon and hype in scientific publishing. A reader will expect all conclusions to be original and of interest, thus successful.

2. The last two paragraph of the Introductory Section- Repairs required.

3. Very less references have been used by authors and can be increased considering the length of the article. To improve the scientific quality of the paper, the following similar and relevant literature is suggested to be added:

1)Meng, X., Leng, X., Shan, C., … & Lu, J. (2023) Vibration fatigue performance improvement in 2024-T351 aluminum alloy by ultrasonic-assisted laser shock peening,

International Journal of Fatigue, 168.

2)Erfan Maleki, Sara Bagherifard, Okan Unal, Alireza Jam, Shuai Shao, Mario Guagliano, Nima Shamsaei. (2023) Superior effects of hybrid laser shock peening and ultrasonic nanocrystalline surface modification on fatigue behavior of additive manufactured AlSi10Mg, Surface and Coatings Technology, 463.

4. Fig.1 suggest to improve the brightness, change the labeling method, similar problems are Fig.2, Fig.3 and Fig.4.

5. English writing format should be noted, the paragraph should be written in front of the top frame.

6. The parts mentioned in the equipment composition described in the second section "Experimental equipment and materials" should be labeled in the figure.

7. The abbreviation "table" is not recommended.

1.The abstract is recommended to be modified to avoid the use of "success" and other similar words. Consider modifying the Abstract “…and successfully completed the processing of 1.206mm inner diameter….”. It could not be made unsuccessfully what you just did and you did not present any debate over its possibility. So, delete the unnecessary word "successfully", as it adds to jargon and hype in scientific publishing. A reader will expect all conclusions to be original and of interest, thus successful.

2. The last two paragraph of the Introductory Section- Repairs required.

3. Very less references have been used by authors and can be increased considering the length of the article. To improve the scientific quality of the paper, the following similar and relevant literature is suggested to be added:

1)Meng, X., Leng, X., Shan, C., … & Lu, J. (2023) Vibration fatigue performance improvement in 2024-T351 aluminum alloy by ultrasonic-assisted laser shock peening,

International Journal of Fatigue, 168.

2)Erfan Maleki, Sara Bagherifard, Okan Unal, Alireza Jam, Shuai Shao, Mario Guagliano, Nima Shamsaei. (2023) Superior effects of hybrid laser shock peening and ultrasonic nanocrystalline surface modification on fatigue behavior of additive manufactured AlSi10Mg, Surface and Coatings Technology, 463.

4. Fig.1 suggest to improve the brightness, change the labeling method, similar problems are Fig.2, Fig.3 and Fig.4.

5. English writing format should be noted, the paragraph should be written in front of the top frame.

6. The parts mentioned in the equipment composition described in the second section "Experimental equipment and materials" should be labeled in the figure.

7. The abbreviation "table" is not recommended.

Author Response

Dear Editors and Reviewers:

I would first like to thank you for your letter and the reviewers’ comments concerning our manuscript. The comments you made were all valuable and helpful for revising and improving our paper and the important guiding significance to our research. We have substantially revised our manuscript after reading the comments provided by the reviewer. I hope to be met with approval. Revised portions are marked in red throughout the paper. Thank you very much for all your help and looking forward to hearing from you soon.

Best regards

Sincerely yours

Yugang Zhao

The main corrections in the paper and the responses to the reviewer’s comments are as follows:

  1. The abstract is recommended to be modified to avoid the use of "success" and other similar words. Consider modifying the Abstract “…and successfully completed the processing of 1.206mm inner diameter….”. It could not be made unsuccessfully what you just did and you did not present any debate over its possibility. So, delete the unnecessary word "successfully", as it adds to jargon and hype in scientific publishing. A reader will expect all conclusions to be original and of interest, thus successful.

Answer:
Thanks for your valuable advice. I have deleted the relevant words.

  1. The last two paragraph of the Introductory Section- Repairs required. 

Answer:
Thanks for your valuable advice. I have finished making revisions to the introduction section, and the revisions are marked in red font.

  1. Very less references have been used by authors and can be increased considering the length of the article. To improve the scientific quality of the paper, the following similar and relevant literature is suggested to be added:

1)Meng, X., Leng, X., Shan, C., … & Lu, J. (2023) Vibration fatigue performance improvement in 2024-T351 aluminum alloy by ultrasonic-assisted laser shock peening,

International Journal of Fatigue, 168.

2)Erfan Maleki, Sara Bagherifard, Okan Unal, Alireza Jam, Shuai Shao, Mario Guagliano, Nima Shamsaei. (2023) Superior effects of hybrid laser shock peening and ultrasonic nanocrystalline surface modification on fatigue behavior of additive manufactured AlSi10Mg, Surface and Coatings Technology, 463.

Answer:
Thanks for your valuable advice. I have added relevant references and revised the introduction section.

  1. Fig.1 suggest to improve the brightness, change the labeling method, similar problems are Fig.2, Fig.3 and Fig.4.

Answer:
Thanks for your valuable advice. I have made changes to the relevant images

  1. English writing format should be noted, the paragraph should be written in front of the top frame.

Answer:
Thanks for your valuable advice. I have revised it.

  1. The parts mentioned in the equipment composition described in the second section "Experimental equipment and materials" should be labeled in the figure.

Answer:
Thanks for your valuable advice. I have modified Figure 1.

  1. The abbreviation "table" is not recommended.

Answer:
Thanks for your valuable advice. I have revised it and marked it as red.

For the revised manuscript, please refer to the attachment. 

Reviewer 2 Report

I wish  the authors success in the development of proccessing technologics using ultrasound. My suggestions  for the article in my review.

Author Response

Dear Editors and Reviewers:

I would first like to thank you for your letter and the reviewers’ comments concerning our manuscript. The comments you made were all valuable and helpful for revising and improving our paper and the important guiding significance to our research. I have revised our manuscript after reading the comments provided by the reviewer. I hope to be met with approval. Revised portions are marked in red throughout the paper. Thank you very much for all your help and looking forward to hearing from you soon.

Best regards

Sincerely yours

Yugang Zhao

The main corrections in the paper and the responses to the reviewer’s comments are as follows:

  1. The article says that the proposed technology allows to reduce the force required to remove the steel rod from the tube after the end of processing. It is necessary to quantitatively show how the efforts have changed when applying the new processing technology. If it is difficult to show this in terms of force or percentage, then at least give a description of the sensations of the operator.

Answer:
Thanks for your valuable advice. The decoring force of the traditional processing method is very high and can only be done by machines, while after ultrasonic processing it can be done manually and the steel core can be decored very easily. I have revised in the manuscript and highlighted them in red.

  1. In fig. 4 shows the position of the forging head in the starting position and the clearance h. If the starting position is understood as the position of the head without oscillation, then a question arises regarding Table 4 and Figure 5. One of the experiments shows h=0.2 mm at an amplitude of 0.186 mm. The question arises: how, with such a small amplitude, such a large gap is overcome.

Answer:
I think there is only one reasonable explanation for this phenomenon: due to the long tube used in the experiment, radial runout and resonance will inevitably occur during the machining process thus causing the ultrasonic vibration head to machine to the electrode surface, and the machining was indeed completed using this set of parameters in the experiment, so I still kept the results of this set of experiments.

  1. In the article, brass is indicated as the material of the tubes. But in output 1 it is talking about a copper tube.

Answer:
Thanks for your valuable advice. In fact the machine can also finish copper tubes, but for the sake of rigor, I will use brass uniformly. The corresponding modifications I have marked in red font.

  1. Table 1 shows that the diameters of the processed pipes included a range of 0.5-2 mm. Section 4 (line 211) refers to an electrode with a diameter of 20 mm.

Answer:
The 20mm in line 211 refers to the sample length of the copper electrode after machining, and I will clearly indicate this in the manuscript, and I will mark the modified parts in red.

  1. If the authors have a sketch or a diagram showing the relationship of the elements of the experimental equipment and the main movements in the processing zone, then it is useful to supplement Figure 1 with this diagram.

Answer:
Thanks for your valuable advice. I have modified Figure 1, I have modified Figure 1, and the specific motion relationships are detailed by the schematic with text.

For the revised manuscript, please refer to the attachment.
